# Inflammatory and immunopathological differences in brains of permissive and non-permissive hosts with *Angiostrongylus cantonensis* infection can be identified using $^{18}$F/FDG/PET-imaging

**Kang-wei Chang[1,2‡], Lian-Chen Wang[3‡], Hung-Yang Wang[4,5], Tzu-Yuan Lin[4,5], Edwin En-Te Hwu[6], Po-Ching Cheng** [4,5,7] *

**1** Neuroscience Research Center, Taipei Medical University, Taipei, Taiwan, **2** Laboratory Animal Center, Taipei Medical University, Taipei, Taiwan, **3** Department of Parasitology, College of Medicine, Chang Gung University, Taoyuan, Taiwan, **4** Department of Molecular Parasitology and Tropical Diseases, School of Medicine, College of Medicine, Taipei Medical University, Taipei, Taiwan, **5** Graduate Institute of Medical Science, College of Medicine, Taipei Medical University, Taipei, Taiwan, **6** The Danish National Research Foundation and Villum Foundation's Center for Intelligent Drug Delivery and Sensing Using Microcontainers and Nanomechanics, Department of Health Technology, Technical University of Denmark, Copenhagen, Denmark, **7** Center for International Tropical Medicine, College of Medicine, Taipei Medical University, Taipei, Taiwan

‡ These authors contributed equally to this work as first authors.
* bonjovi@tmu.edu.tw

## Abstract

### Background

*Angiostrongylus cantonensis* is a parasite that mainly infects the heart and pulmonary arteries of rats and causes human eosinophilic meningitis or meningoencephalitis in certain geographical areas. Current diagnostic methods include detection of the parasite in cerebrospinal fluid (CSF) and eosinophilic immune examination after lumbar puncture, which may be risky and produce false-positive results. $^{18}$F- Fluorodeoxyglucose (FDG), a Positron emission tomography (PET) tracer, has been used to assess different pathological or inflammatory changes in the brains of patients. In this study, we hypothesized that *A. cantonensis* infection-induced inflammatory and immunomodulatory factors of eosinophils result in localized pathological changes in the brains of non-permissive hosts, which could be analyzed using *in vivo* $^{18}$F-FDG PET imaging.

### Methodology/Findings

Non-permissive host ICR mice and permissive host SD rats were infected with *A. cantonensis*, and the effects of the resulting inflammation on $^{18}$F-FDG uptake were characterized using PET imaging. We also quantitatively measured the distributed uptake values of different brain regions to build an evaluated imaging model of localized neuropathological damage caused by eosinophilic inflammation. Our results showed that the uptake of $^{18}$F-FDG increased in the cerebellum, brainstem, and limbic system of mice at three weeks post-

**Data Availability Statement:** All relevant data are within the manuscript and its Supporting Information files.

**Funding:** This study was supported by grant MOST 104-2320-B-038-059 to PCC from the Ministry of Science and Technology, Taipei, Taiwan; and in part by grants: the Villum Foundation (Grant No. 9301 to EETH) for Intelligent Drug Delivery and Sensing Using Microcontainers and Nanomechanics (IDUN); and the LEO Foundation (LF-OC-20-000370 to EETH) project. The funders had no role in study design, data collection and analysis, decision to publish, or preparation of the manuscript.

**Competing interests:** The authors have declared that no competing interests exist.

infection, whereas the uptake in the rat brain was not significant. Immunohistochemical staining and western blotting revealed that Iba-1, a microglia-specific marker, significantly increased in the hippocampus and its surrounding area in mice after three weeks of infection, and then became pronounced after four weeks of infection; while YM-1, an eosinophilic chemotactic factor, in the hippocampus and midbrain, increased significantly from two weeks post-infection, sharply escalated after three weeks of infection, and peaked after four weeks of infection. Cytometric bead array (CBA) analysis revealed that the expression of TNF in the serum of mice increased concomitantly with the prolongation of infection duration. Furthermore, IFN-γ and IL-4 in rat serum were significantly higher than in mouse serum at two weeks post-infection, indicating significantly different immune responses in the brains of rats and mice. We suggest that [18]F-FDG uptake in the host brain may be attributed to the accumulation of large numbers of immune cells, especially the metabolic burst of activated eosinophils, which are attracted to and induced by activated microglia in the brain.

## Conclusions

An *in vivo* [18]F-FDG/PET imaging model can be used to evaluate live neuroinflammatory pathological changes in the brains of *A. cantonensis*-infected mice and rats.

## Author summary

*Angiostrongylus cantonensis* infection in humans can cause brain inflammation, and there is currently no effective diagnostic method to clarify the pathological changes in the brain. Current diagnostic methods include CSF detection and eosinophilia immune examination after lumbar puncture, but these protocols are risky and may produce false-positive responses. Non-invasive medical imaging techniques might allow for the circumvention of these problems. The effective use of [18]F-FDG PET imaging as a diagnostic tool for parasitic infections has been reported. In this study, we hypothesize that the *A. cantonensis* infection-induced inflammatory and immunomodulatory factors of eosinophils have an impact on the localized pathological changes in the brains of non-permissive hosts. These changes can be analyzed using live *in vivo* [18]F -FDG PET medical imaging. *A. Cantonensis* was infected in non-permissive ICR mice and permissive SD rats, after which the effects of the resulting inflammation on energy metabolism were characterized via nuclear medicine imaging. We also quantitatively measured the distributed uptake values of different brain regions, to build an evaluated imaging model of localized neuropathological damages caused by eosinophil-inflammation. Through the completion of this study, we intend to develop a specific radiological medical imaging analysis model of angiostrongyliasis for important endemic zoonotic parasitic diseases. We are trying to develop a cross-disciplinary research project combining parasitology and nuclear medicine and to establish an imaging model for analyzing angiostrongyliasis that can be applied to **clinical evaluation and prognosis.**

## Introduction

*Angiostrongylus cantonensis* is a parasite that mainly infects the heart and pulmonary arteries of rats. However, it also causes human eosinophilic meningitis and meningoencephalitis in the Far East, Southeast Asia, and Pacific islands [1–3]. When mammals ingest the intermediate hosts, such as terrestrial or freshwater mollusks, they become infected with the parasite [4–6].

After infecting rats, third-stage larvae migrate from the intestines to the brain and spinal cord, where they develop into fourth- or fifth-stage larvae. Subsequently, they migrate to the heart and pulmonary arteries and mature into adults [7]. In contrast, larvae remain in the brains of non-permissive hosts such as mice and humans, and their development is arrested at the fourth or fifth larval stage [8]. *A. cantonensis* is an important zoonotic parasite found in local communities in Taiwan. Two recent outbreaks of infection caused by *A. cantonensis* were reported in 1999 and 2001 [9,10]. The presence of larvae in the brains of human patients causes brain and spinal cord symptoms such as headaches, fever, vomiting, lethargy, neck stiffness, liver and spleen enlargement, and increased cerebrospinal fluid (CSF) pressure [3,11].

It must be noted that infection caused by *A. cantonensis* in different host species triggers different levels of immune responses [12]. The presence of *A. cantonensis* in the brains of non-permissive hosts induces eosinophilia in the CSF [13–15], and significantly increases the levels of both systemic and local Th2 cytokines [12,16,17]. The concentrations of antibodies active against the parasite also vary in the serum and CSF of patients with eosinophilic meningitis infected with *A. cantonensis* [9,10,18]. However, the process by which *A. cantonensis* infects hosts of different genotypes and activates diverse local and systemic immune mechanisms that trigger pathological lesions in the brain remains poorly understood. Central nervous system (CNS) injuries caused by *A. cantonensis* include hemorrhage, vascular dilatation, focal necrosis with neuronal loss, and infiltration of inflammatory cells [12,19]. Cerebral pathogenesis is characterized by infiltration of eosinophils that induce significant inflammatory reactions in response to the presence of immature parasites in the nervous system [19,20]. Eosinophils are recruited and several immune-mediated cytotoxic attacks are initiated to eliminate the pathogen, resulting in severe tissue damage [21–23]. The differences in innate immune responses to parasitic infections are largely due to the distinct key immune cells that are active in the host during invasion and development. Microglia may play a role in immune modulation during *A. cantonensis*-induced brain inflammation [22]. Ym1 is an eosinophil chemotactic factor secreted by alternatively activated macrophage-derived microglia that participates in eosinophilic chemotactic inflammation in the CNS [24]. Therefore, microglia are activated to mediate blood-brain barrier dysfunction and enhance eosinophilia and type-2 immune responses [25,26]. Although several inflammatory mediators are involved in *A. cantonensis* infection, the mechanisms through which these mediators influence the pathogenicity and pathophysiology of cerebral angiostrongyliasis remain unclear.

Current diagnostic methods include the detection of the parasite in CSF and eosinophilic immune examination after lumbar puncture; however, these protocols are risky and may produce false-positive results [27,28]. Non-invasive medical imaging techniques may help circumvent these problems. Shyu et al. characterized angiostrongyliasis in the brains of rats infected with *A. cantonensis* larvae using magnetic resonance imaging (MRI) [28]. Positron emission tomography (PET) is a functional medical imaging technique that can provide a specific diagnosis earlier than conventional morphological imaging techniques such as radiography, computed tomography (CT), MRI, and other anatomical imaging techniques [29–31]. PET tracers, such as [18]F-FDG, can be used to observe different pathological or physiological changes in patients. Glucose metabolism is more active in regions of cell proliferation and rapid energy consumption, such as tumors, inflammation, and granulomas, compared to normal tissues. This implies that in such regions, [18]F-FDG is absorbed more abundantly, and regions of cell proliferation appear as bright, highlighted regions in the PET images [31–33]. The effective use of [18]F-FDG PET imaging as a diagnostic tool for parasitic infections has been reported for alveolar echinococcosis [34,35] and bile-chemotactic migration of *Clonorchis sinensis* [36]. However, research on the pathogenicity of parasitic infections is still lacking and is worthy of further exploration.

In this study, we hypothesized that the *A. cantonensis* infection-induced inflammatory and immunomodulatory factors in eosinophils have an impact on localized pathological changes in the brains of non-permissive hosts, and these changes can be analyzed using *in vivo* [18]F-FDG PET imaging. Non-permissive ICR mice and permissive SD rats were infected with *A. cantonensis*, and the effects of the resulting eosinophilic inflammation on energy metabolism were characterized using live nuclear imaging. We also quantitatively measured the distributed uptake values of different brain regions to build an evaluated imaging model of localized neuropathological damages caused by eosinophilic inflammation. We aimed to develop a cross-disciplinary research project combining parasitology and nuclear medicine, and establish an imaging model for the analysis of angiostrongyliasis that can be applied to **clinical evaluation and prognosis.**

## Materials and methods

### Ethics statement

Experiments were carried out under humane conditions with approval (license number:LAC-2014-0248) from the Institutional Animal Care and Use Committee (IACUC) of Taipei Medical University, and conducted in accordance with the NIH Guide for the Care and Use of Laboratory Animals [37].

### Animals

Six- and five-week-old male outbred ICR mice and SD rats were both be purchased from Bio-LASCO Taiwan Co., Ltd, Taipei, Taiwan. Animal

### Parasite and infection

The third-stage *A. cantonensis* larvae (L3) were provided by the laboratory of Prof. Lian-Chen Wang, Department of Parasitology, Chang Gung University. *A. cantonensis* was maintained by serial passage in *Biomphalaria glabratus*, and in SD rats in the laboratory. Infective larvae were obtained by 0.6% pepsin digestion (pH 2, 37˚C, 1 hr) of snails previously infected with the first-stage larvae. **ICR mice and SD rats were orally infected with 50 and 100 L3, respectively, as in our previous publication [38,39].** The infected L3 mice and rats were divided into experimental groups of six and four, respectively. For each group, PET imaging was performed at two and four weeks after infection. After imaging, they were euthanized, and the larvae were counted. Brain tissue specimens were collected at this time. Normal uninfected mice and rats served as the control groups.

### Radiopharmaceutical

2-Deoxy-2-[[18]F] fluoroglucose, was purchased from PET Pharm Biotech Co., Ltd. using a modified version of the method as described [40]. The purity of [[18]F] FDG was > 95% as assessed by radio-HPLC and radio-TLC. The prepared [[18]F] FDG was immediately sent to the animal imaging center for NanoPET imaging and *ex vivo* autoradiography in experiment animals.

### NanoPET image acquisition

Acquisitions were made on a nanoPET SuperArgus system (Sedecal, Madrid, Spain) which had an effective axial/transaxial FOV of 4.8/6.7 cm, a spatial resolution of less than 2 mm, and a sensitivity above 2.5% in the whole FOV. Each scan was corrected for randoms, scatter, and attenuation, and the images were reconstructed using a 2D OSEM algorithm (GE Healthcare,

France) into voxels of 0.3875 x 0.3875 x 0.775 mm3. For PET-based monitoring of brain region capability, radiotracers ([18F] FDG) were administered as a bolus (100–200 μL) in normal saline at an activity of approximately 18.5–37 MBq by rapid manual tail vein injection. The experimental animals were placed in a stereotactic head holder in a prone position on the bed of the scanner. Animals were anesthetized with isoflurane gas (3% isoflurane in 50% oxygen, 1 mL/min). For tracer imaging evaluation, PET images were acquired for post-injection brain neuroimaging as static images from 120 to 130 min.

## Data analysis of PET imaging

Reconstructed PET data were further processed with PMOD 3.3 software (PMOD Technologies Ltd., Zürich, Switzerland), co-registered to the MRI template, and the defined regions of interest (ROIs) were drawn. The Pmod v3.3 mice and rat brain regions template were used to define the regions of interest (ROI). We set up a PMOD template for the brain including the striatum, cortex, hippocampus, thalamus, cerebellum, hypothalamus, amygdala and Brain stem respectively. The relative ratio of SUV (SUVr) of the target regions was calculated by the mean counts per pixel in different brain regions and divided by the amygdala (background, BG), that is, ST/BG. The task was carried out by an experienced technologist to avoid inter- and intra-reader variability in ROI analysis.

## Worm recovery and sample collection

After PET imaging, the experimental and control groups of rats and mice were weighed before being euthanized. Serum samples were collected from the animals via heart puncture, and kept overnight at 4˚C, before being centrifuged at 15,000 × g for 10 min. Then, the supernatant were collected and stored at -70˚C. After the collection of serum, the brains of the host were moved to a Petri dish under a dissecting microscope, and L5 larvae or adults were counted. Then, the brain tissue was divided into three parts, and one was fixed in formalin in preparation for the histopathological staining. The remaining two parts were respectively used for real-time RT-PCR and western blotting experiments. Additionally, the hearts and lungs of rats were also recovered for adult worm counting.

## Histopathological examination

Half of the rats and mice brain tissue specimens (>0.5 cm3) were fixed in 10% neutral formalin for 24 hr. The fixed specimens were embedded in paraffin and cut into 5-μm-thick sections. Tissue specimens from uninfected hosts were taken from similar locations in the brain to serve as controls. These sections were stained with hematoxylin-eosin and observed under a microscope for inflammatory responses and pathological damage. **The eosinophils were quantified according to a previously described procedure for assessing histopathology with the aid of computer image analysis software (Image J, NIH, USA). This program is available for free on the Internet (at http://rsb.info.nih.gov/ij/index.html) [41].**

## Immunohistochemistry staining

For immunohistochemical analysis, brain sections were deparaffinized in xylene and rehydrated in alcohol. After treatment with a peroxidase-blocking agent, sections were incubated with biotin-labeled anti-YM-1 and anti-Iba-1 antibodies (R&D Systems, Minneapolis, MN). They were then be kept overnight at 4˚C, and subsequently manipulated as described by the manufacturer of the Avidin Biotinylated enzyme complex (Vectastain R ABC reagent, Vector Laboratories, Burlingame, CA, USA), with diaminobenzidine (DAB) staining for microscopic

observation. To assess the number of positive expression cells present in the stained tissue sections, ten random fields were captured at 200× magnification for each group of mice and rats. The positive cells were quantified according to a previously described procedure for assessing histopathology with the aid of computer image analysis software (Image J, NIH, USA). This program is available for free on the Internet (at http://rsb.info.nih.gov/ij/index.html) [41].

## Preparation of brain RNA and protein

The preparation of RNA for real-time RT-PCR experiments was extracted from the brain tissue using TRIzol Reagent (Invitrogen, USA), and then using an RNeasy Mini Kit (Qiagen, Germany). Purified RNA was quantified at OD260/280 nm using an ND-1000 spectrophotometer (Nanodrop Technology, USA). The purity of all measured RNA specimens was above a ratio of 2.0, and all RNA integrity number (RIN) values were above 7 in the experiment. Another brain tissue sample was added RIPA Protein lysis buffer (Pierce, Rockford, IL) and then homogenized in 0.15 M PBS (pH 7.4) in a glass tissue grinder at 4˚C, and sonicated in an ultrasonic disintegrator (Soniprep 150, MSE Scientific Instruments, Manor Royal, UK), then placed in a -70˚C refrigerator until subsequent experiments using western blotting tests.

## Isolation of RNA, cDNA synthesis, and real-time reverse transcriptase-polymerase chain reaction (real-time RT-PCR)

Total RNA was reverse transcribed using the Moloney Murine Leukemia Virus (MMLV) reverse transcriptase (Promega, Madison, WI, USA) to generate cDNA. Each cDNA pool was stored at -20˚C until real-time PCR analysis was completed. Primer pair specificity was validated by performing RT-PCR using common reference RNA (Stratagene, La Jolla, CA, USA) as a DNA template. The primers used were shown in Table 1, and Glyceraldehyde-3-phosphate dehydrogenase (GAPDH) and 18s were used as endogenous reference genes. Real-time PCR was performed using the lightcycler nano real-time PCR system (Roche Diagnostics, Mannheim, Germany) using LightCycler 480 SYBR Green I Master (Roche Diagnostics). Briefly, 10 μl reactions were utilized, containing 2 μl of Master Mix, 2 μl of 0.75 μM forward primer and reverse primer, and 6 μl of the cDNA sample. Each sample was run in triplicate. The real-time PCR program included 3 min at 95˚C; 45 cycles of 10 s durations at 95˚C, and 30 s at 60˚C. Data analysis was performed using the LightCycler Nano software version 1.0 (Roche). The gene was considered differentially expressed only when there was an upregulation of >2-fold or a downregulation of <0.5-fold.

**Table 1. Worm burden in mice and rats infected with *A. cantonensis*.**

| Species | Infected weeks | Number of hosts | Infected region | Worm recovery |
|---|---|---|---|---|
| SD rats | 2w | 4 | Brain | 12.75±1.80 |
| | | | Heart | 0.25±0.50 |
| | 3w | 4 | Brain | 15.00±1.95 |
| | | | Heart | 0.40±0.89 |
| | 4w | 4 | Brain | 1.00±0.12 |
| | | | Heart | 18.00±2.31 |
| ICR mice | 2w | 6 | Brain | 6.67±3.46 |
| | 3w | 6 | Brain | 13.83±7.35 |
| | 4w | 6 | Brain | 4.50±5.23 |

Data are expressed as mean ± SD. None of the ICR mice infected with 50 larvae died due to infection.

### Western blot analysis of specific protein expression

Homogenized brain cell proteins separated by 13.5% homologous SDS-PAGE were electrophoretically transferred to a polyvinylidene difluoride (PVDF, MSI, Westborough, US) membrane. The membranes were blocked with 5% BSA (Bio-Rad) in PBS for 30 min at 37°C, and then incubated with diluted 1° Ab at 37°C for 1 h. After washing, membranes were incubated with HRP-conjugated 2° Ab at 37°C for 0.5 h. Protein expression was detected using an ECL-Plus Western Blot Detection system (GE Healthcare UK Ltd.) according to the manufacturer's instructions. All experiments were performed at least thrice. Graphical analysis of band density was performed using ImageJ software (version 1.41o) (National Institutes of Health, Bethesda, MD, USA) (http://rsb.info.nih.gov/ij/). We used anti-YM-1, and Iba-1 antibodies purchased from R & D Systems (Minneapolis, MN), and the BD Biosciences (Billerica, MA) company.

### Cytokine ELISA

The cell culture medium was centrifuged (13,000 ×$g$, 20 min, 4°C), and the sera of the mice and rats were assessed using a mouse and rat inflammatory cytokine CBA (BD Biosciences, San Diego, USA) for the cytokines IFN-$\gamma$, IL-2, IL-4 and tumor necrosis factor-alpha (TNF-$\alpha$). IL-5 in mouse serum was performed separately. The cytokine capture bead, PE detection reagent, and recombinant standards or test samples were incubated for 3 h at room temperature. FACSCanto flow cytometer (BD Biosciences, San Diego, USA) was used to acquire the data, which were analyzed using the BD CBA analysis software to produce graphs [42,43].

### Statistical analysis

All sample levels were compared using one-way ANOVA of SPSS 18.0 software (SPSS Inc.; Chicago, IL, USA). Results between individual groups were analyzed using two-tailed Student's *t*-tests. In all analyses, $p < 0.05$ was considered significant and data were expressed as the means ± standard deviations (SD).

## Results

### Worm burden in mice and rats infected with *A. cantonensis*

Mice and rats were infected with third-stage larvae and euthanized. Their larval numbers were counted at 2, 3, and 4 weeks post-infection. As shown in Table 1, the infected region of rats significantly changed from the brain in the second week to the heart in the fourth week. The infection worm burdens in SD rats during 2–4 weeks were 12.75±1.80, 15.00±1.95, and 1.00 ±0.12 in brain and 0.25±0.50, 0.4±0.89, and 18.00±2.31 in heart, respectively. Contrarily, the number of infected worms in the brains of ICR mice from the second week to peak in the third week and declined at the fourth week were 6.67±3.46, 13.83±7.35 and 4.50±5.23, respectively. **Meanwhile, the worms were not detected in the hearts of ICR mice for all.**

### [18]F-FDG PET /CT imaging analysis of different brain regions of infected mice and rats

Mice and rats were infected with 50 and 100 third-stage larvae of *A. cantonensis*, respectively. After two–to four weeks post-infection, both infected and uninfected control mice and rats were administered approximately 18.5 MBq and 37 MBq [18]F-FDG via tail vein injection. All animals were continuously scanned using an *in vivo* nanoScan PET/CT. Data for different brain regions in normal and infected animals were acquired in a list mode and quantified by calculating the mean standard uptake value (SUV) of each delineated region of interest (ROI).

As depicted in Fig 1, radioactivity uptake in different brain regions of both mice (A) and rats (B) are displayed for each group at different time points post-infection. In mice, increased uptake was observed at two weeks post-infection in a majority of the regions of the brain, such as the cerebellum and brainstem. It significantly increased at three weeks post-infection, followed by a decrease at four weeks post-infection. Conversely, radioactivity uptake in rats did not exhibit a significant trend during infection, except in the cortex and cerebellum, which showed a slight increase at three- and four weeks post-infection. Representative PET images of the mouse and rat brains are shown in (C) and (D), respectively. Specific brain areas were delineated for further analysis. ROI analysis (C) of mice showed that infected mice exhibited significantly increased [18]F-FDG uptake in the hippocampus, thalamus, cerebellum, and brainstem at three weeks post-infection in comparison to normal mice (** $p < 0.01$, *** $p < 0.001$, **** $p < 0.0001$), especially in the cerebellum, where it increased from two weeks and

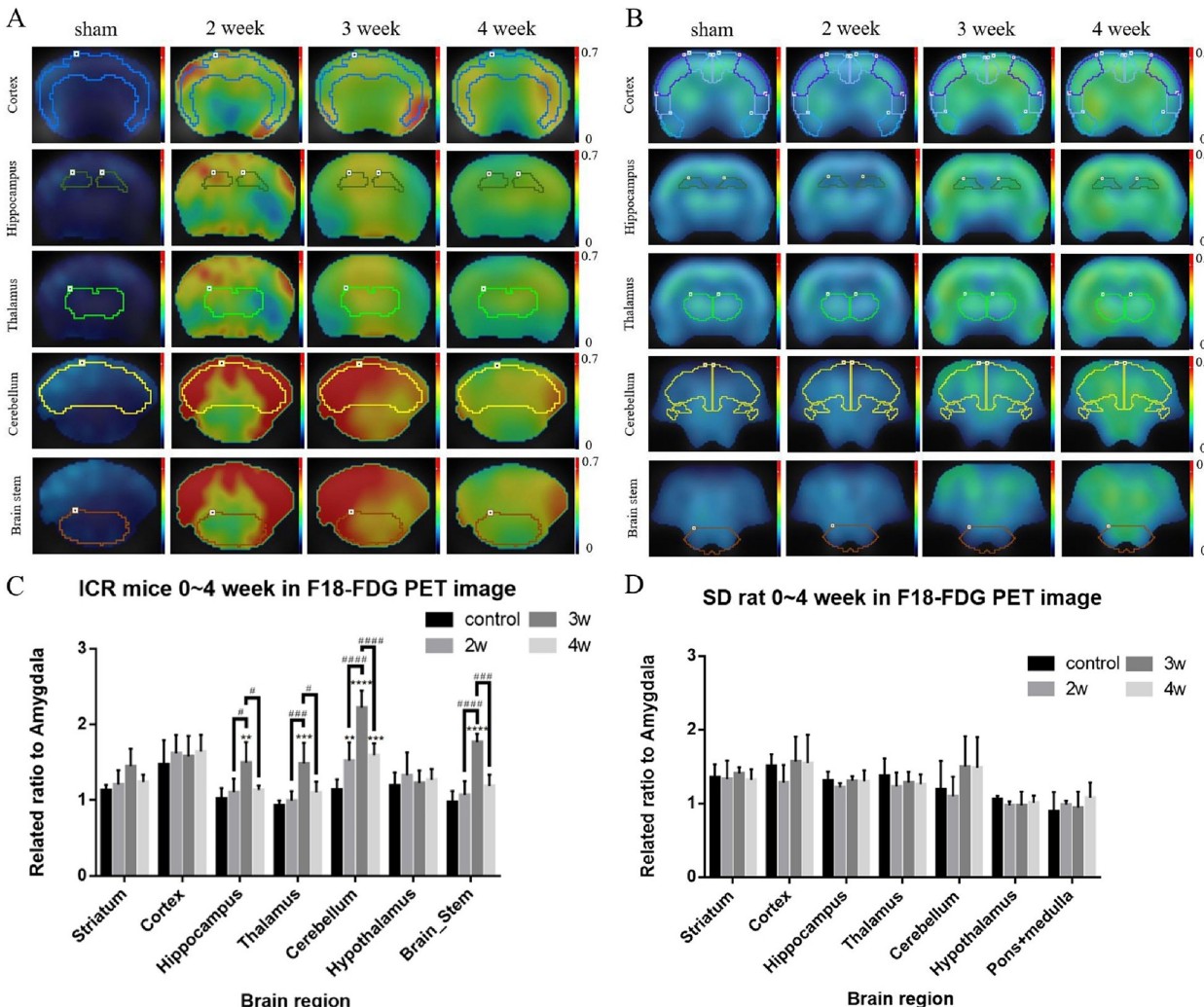

**Fig 1.** Images (transaxial view) of fused FDG-PET/CT slices from normal mice (A) and rats (B), or mice and rats infected with *A. cantonensis* 2–4 weeks after 120 min of [18]F-FDG injection. The color bar indicates standardized uptake value (SUV). **The image shown is representative of a typical result.** (C) (D) The special brain areas as indicated were circled in the figure for analysis. Data are expressed as mean + standard deviation. **$p < 0.01$, *** $p < 0.001$, **** $p < 0.0001$ when compared with the normal control. #$p < 0.05$, ###$p < 0.001$, ####$p < 0.0001$ indicate comparisons between two indicated groups.

continued until four weeks. In addition, in the hippocampus, thalamus, cerebellum, and brainstem, there was a peak in uptake at three weeks post-infection, which was significantly higher than that at two weeks post-infection, followed by a subsequent decrease at four weeks (### $p < 0.001$, #### $p < 0.0001$). In contrast, in the rat brain, there were no obvious changes in any region. Only the cortex and cerebellum showed a slight increase in radioactivity uptake at three and four weeks after infection, however, the difference was not significant. These findings suggest that ¹⁸F -FDG PET imaging can be used to discern differences in different brain regions between mice and rats during infection. This implies that ¹⁸F -FDG PET imaging can serve as a feasible evaluation tool for angiostrongyliasis models.

## Pathological changes and histological analysis of microglia and eosinophils in the brain of mice and rats during *A. cantonensis* infection

We assessed the histopathological changes in the brains of mice and rats following *A. cantonensis* infection. As depicted in Fig 2, pathological examination of normal control mice and rats revealed no inflammatory cells in the brain tissue. However, after *A. cantonensis* infection, there was an increase in the accumulation of immune and inflammatory cells in the brain regions located between the cortex and midbrain, as well as between the hippocampus and thalamus, in mice at two weeks post-infection (**Fig 2B; p<0.05**). We found an increased accumulation of eosinophils around the limbic system and midbrain at three weeks post-infection (**Fig 2B; p<0.01**). This inflammatory infiltration gradually intensified and continued to develop until four weeks post-infection (**Fig 2B; p<0.0001**). **In contrast, in rats, inflammatory infiltration between the hippocampus and thalamus only slightly appeared at four weeks post-infection, which was comparatively less than in the mice (Fig 2B; p < 0.05, < 0.001, < 0.0001).** The infiltration induced by *A. cantonensis* infection may also be linked to the activation of immune cells, especially eosinophils, in tissues surrounding the limbic system.

The values are expressed as means ± SD, *p < 0.05, **p < 0.01, ****p < 0.0001 when compared with the normal control. #p < 0.05, ###p < 0.001, ####p < 0.0001 indicate comparisons between two indicated groups. Slides were visualized under a microscope (Leica, Bensheim, Germany) at (1) 40× and (2) 100× magnification.

To validate the relationship between microglial activation and inflammation, we assessed the expression of the microglia-specific marker, Iba1, in the rodent brain. Brain Iba1 expression levels were evaluated using immunohistochemically stained sections obtained from both infected and normal mice and rats (Fig 3A). Expression levels were significantly enhanced in the brain parenchymal tissues of *A. cantonensis*-infected mice, particularly in the regions surrounding the limbic system. **Iba-1 expression exhibited a significant increase in the hippocampus and its surrounding area in mice at three weeks post-infection in comparison to the control, and this increase was even more pronounced after four weeks of infection (Fig 3A3; p<0.0001). Interestingly, in infected rats, Iba-1 expression did not increase significantly, except in the hippocampus of brain tissues, particularly in rats at three- and four weeks post-infection (Fig 3A3; p<0.05 and p<0.0001, respectively, compared to mice).** On the other hand, to validate the relationship between eosinophil activation and inflammation, we examined the expression of the eosinophil chemotactic factor, YM-1, in the brains of mice and rats (Fig 3B). YM-1 expression was less evident in the brain parenchymal tissues of both normal mice and rats. However, as the infection progressed, the levels of YM-1 expression in the hippocampus and midbrain increased significantly from two weeks post-infection, sharply escalated after three weeks of infection, and peaked after four weeks of infection (Fig 3B3; p<0.01, p <0.0001, p <0.0001, as compared to control). In contrast, similar

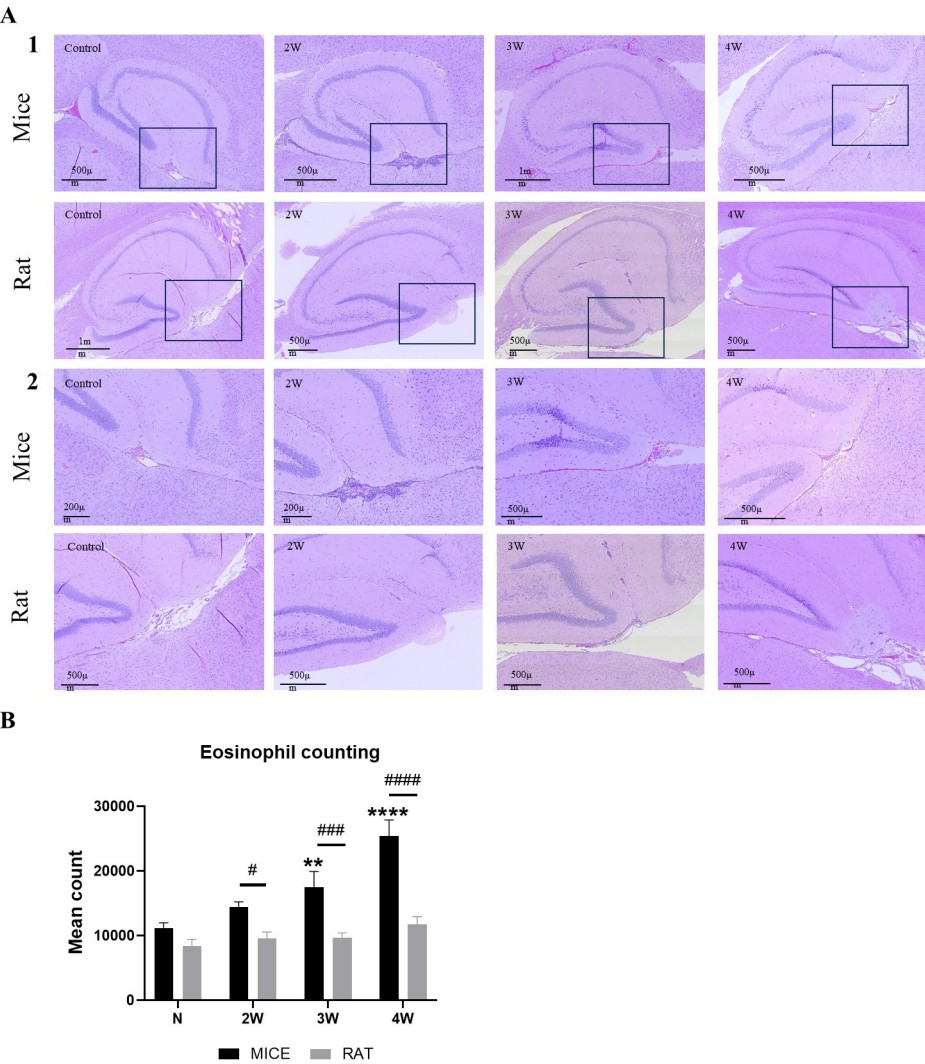

**Fig 2. Histological changes in the brains of normal controls and infected animals at 2–4 weeks. (A1) H&E staining of brain tissue sections of normal mice and rats and those infected with *A. cantonensis* after 2–4 weeks is shown. (A2) The magnified image from 2-4-weeks post-infection and normal mice and rats. (B) Quantitative analysis of eosinophil numbers.** After 2 weeks of infection, increased numbers of immune and inflammatory cells appeared in the mouse brain between the hippocampus, midbrain, and thalamus. Inflammatory cell infiltration further increased at 4 weeks post-infection. However, inflammatory infiltration between the hippocampus and thalamus of rats was delayed until 4 weeks post-infection.

results were not observed in infected rats, with only a lower increase in YM-1 expression around the hippocampus at three- and four weeks post-infection (p<0.0001). Moreover, YM-1 expression was not as pronounced in rats as in mice during all the infections (Fig 3B3; p<0.05, <0.0001, <0.0001, as compared to the mice at two, three- and four weeks post-infection, respectively).

## Detection of Iba1 and YM-1 expression in the brains of mice and rats with *A. cantonensis* infection

To verify the actual changes in Iba-1 and YM-1 levels in the brains of mice and rats after *A. cantonensis* infection, qPCR was used to detect the protein expression levels in the brains at

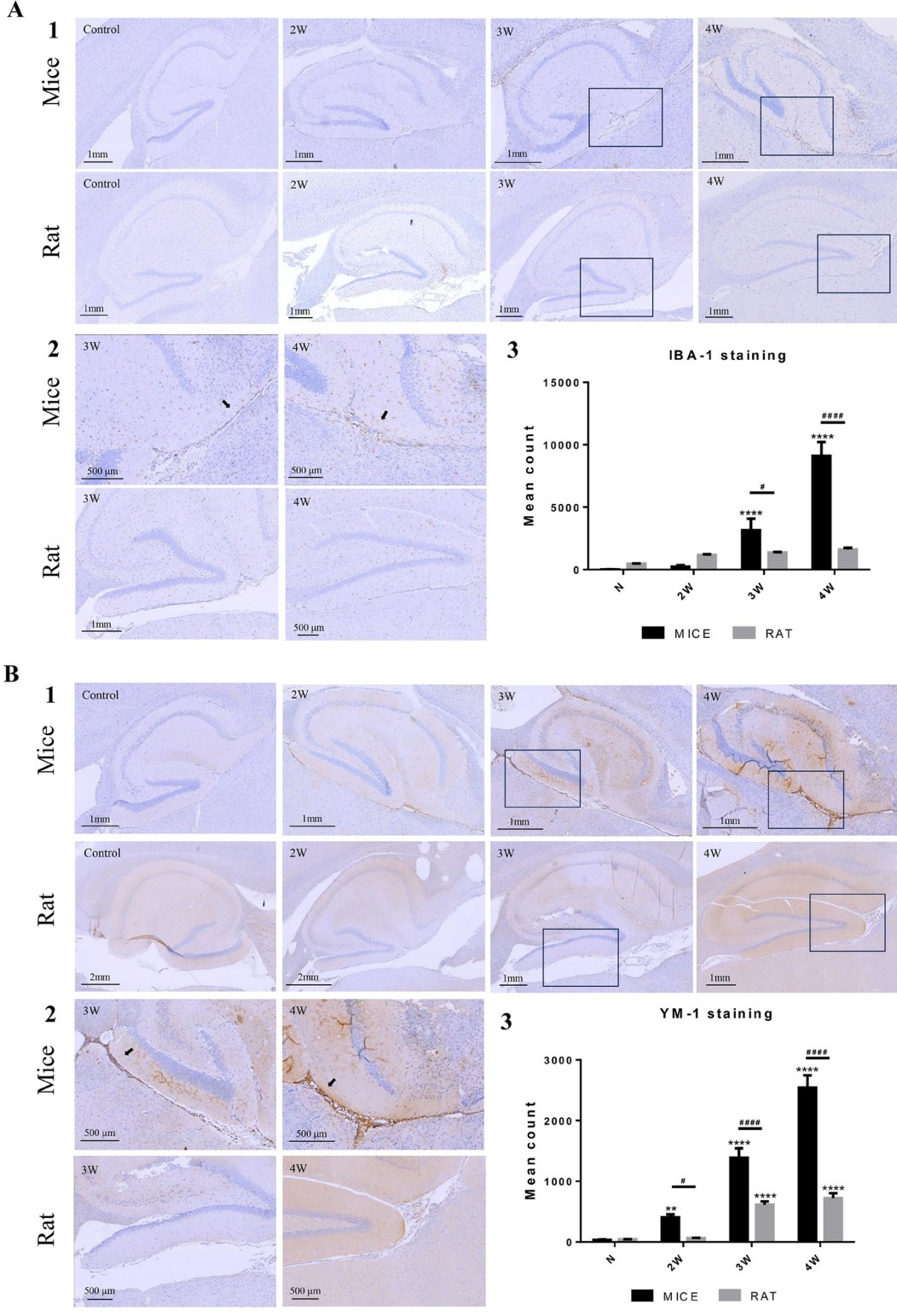

**Fig 3.** Immunohistochemical staining for microglial and eosinophilic chemotactic factors in the brains of infected mice and rats (A1) Microglial IHC staining of mouse and rat brain sections at 2–4 weeks post-infection and normal control. (A2) The magnified image from 3- and 4-weeks post-infection, where the arrow indicates microglia-accumulated signals. (A3) Quantitative analysis of IBA-1 signals. (B1) IHC staining for eosinophil chemotactic factor of mouse and rat brain sections at 2–4 weeks post-infection, and normal control. (B2) The magnified image of 3- and 4 weeks post-infection, where the arrow

indicates eosinophil-accumulated signals. (B3) Quantitative analysis of YM-1 signals. The values are expressed as means ± SD, **p < 0.01, ****p < 0.0001 when compared with the normal control. #p < 0.05, ####p < 0.0001 indicate comparisons between two indicated groups. Slides were visualized under a microscope (Leica, Bensheim, Germany) at (1) 40× and (2) 100× magnification.

different time points after infection. Fig 4A shows that Iba-1 gene expression in the mouse brain began to increase sharply at two weeks post-infection, then decreased in the third week (p<0.001 compared with two weeks), and rebounded slightly at four weeks post-infection (p < 0.01 compared with two weeks). However, the expression levels during infection were significantly higher in comparison to the control group (p < 0.05, p <0.0001, and p <0.0001 compared with the control group). The expression of Ym-1 in the mouse brain increased at two weeks post-infection and gradually increased in the third week, reaching a significant level in the fourth week [(p < 0.05, p < 0.0001 compared to the control); (p < 0.01, p <0.0001 compared with two weeks). On the other hand, the expression of Iba-1 and YM-1 genes in the brain of rats was not as significant as that in mice, and no change was observed at two, three, and four weeks post-infection. At the transcriptional level, expression of Iba-1 (Fig 4B) in the mouse brain was significantly increased at two-week post-infection (p < 0.01), which decreased at three-week post-infection and rebounded at four weeks post-infection (p < 0.01, <0.05 compared with two weeks). The expression of Ym-1+Ym-2 in the mouse brain also increased in the third week (p < 0.01) and then significantly increased in the fourth week (p < 0.0001 compared to control; p<0.01 and p <0.05, compared to two and three weeks, respectively), whereas in rats, the expression of Ym-1+Ym-2 protein in the brain was only slightly increased at four-week post-infection (Fig 4B; p <0.05, compared to two and three weeks, respectively). Regardless of mice or rats, Iba-1 and YM-1 protein and gene expression trends in the brain were similar during *A. cantonensis* infection.

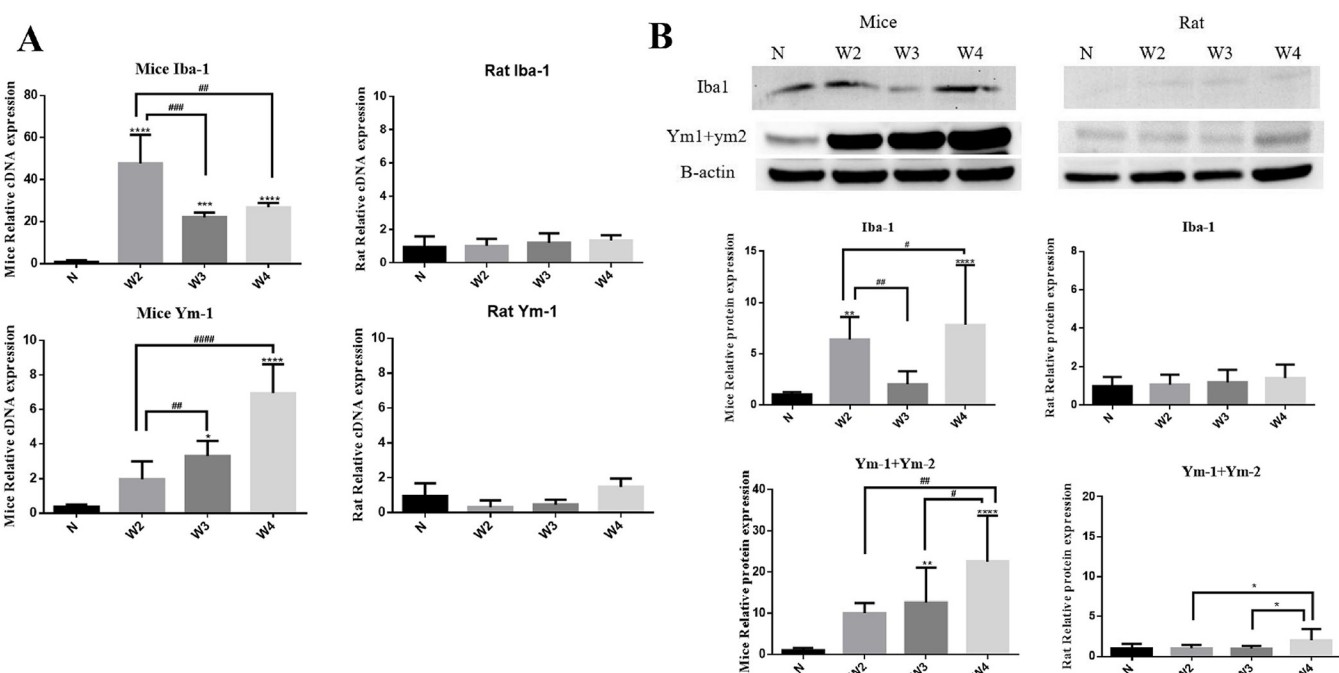

**Fig 4. qPCR and western blot analyses of IBA, YM-1, and YM-1+YM-2 in the brain.** (A) Relative expression of IBA-1 and YM-1 in mice and rats. (B) Relative expression of IBA-1 and YM-1+YM-2 in mice and rats. The values are presented as means ± SD, *p < 0.05, **p < 0.01, ***p < 0.001, ****p < 0.0001 when compared with the normal control. #p < 0.05, ##p < 0.01, ###p < 0.001, ####p < 0.0001 indicate comparisons between two indicated groups.

## Analysis of related cytokines in serum during *A. cantonensis* infection

To further confirm the effect of macrophage-induced chemotaxis after *A. cantonensis* infection on subsequent host immune phenotypes, cytokines in mouse and rat sera were tested at two, three- and four weeks post-infection. As shown in Fig 5A, mouse TNF increased at two weeks post-infection, was strongly expressed at three weeks post-infection, and reached a peak at four weeks post-infection ($p < 0.05$). On the other hand, in rats, it was only slightly expressed at two weeks post-infection, and the level was much lower in comparison to that in mice ($p < 0.05$). The expression of IFN-γ in mice rose slowly after infection until it reached the peak at four weeks post-infection; while in rats, IFN-γ sharply increased at two weeks post-infection ($p < 0.01$ as compared to mice) and then declined at three weeks post-infection. Among the type-2 cytokines, IL-4 levels in mice slowly increased after infection and peaked at four weeks post-infection. However, rats showed a large amount of IL-4 at two weeks post-infection ($p < 0.001$, as compared to mice), which declined after three weeks, but was reversed at four weeks post-infection ($p < 0.0001$ as compared to mice). IL-2 expression in mice showed only a slight increase at three weeks post-infection, while rat IL-2 expression increased from two to

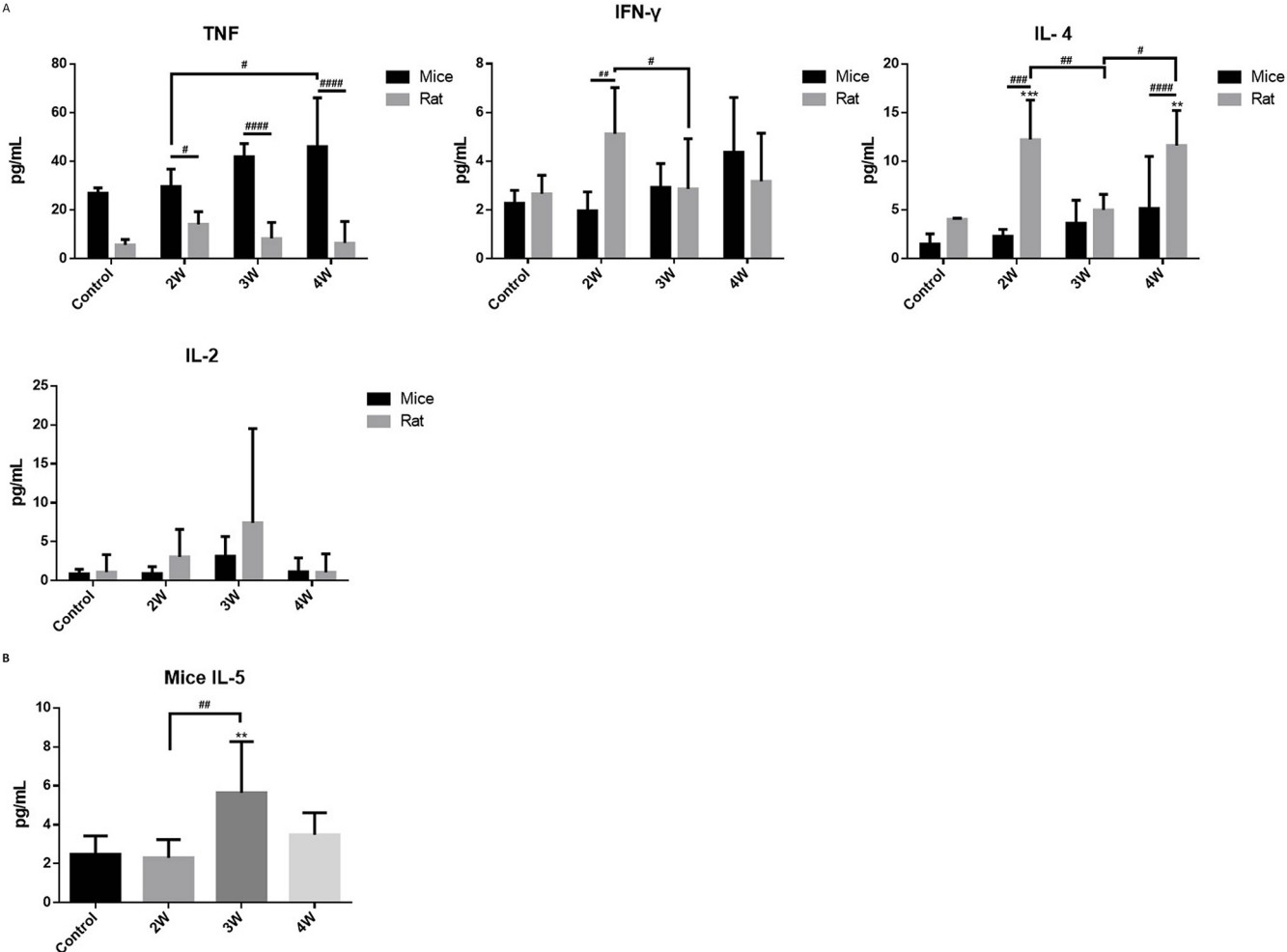

**Fig 5. The cytokine ELISA analysis of mice and rats for IFN-γ, IL-2, IL-4, IL-5, and tumor necrosis factor-alpha (TNF-α).** (A) Mouse and rat cytokine levels. (B) IL-5 levels in mice. The values are presented as means ± SD, *$p < 0.05$, **$p < 0.01$, ***$p < 0.001$, ****$p < 0.0001$ when compared with the normal control, #$p < 0.05$, ##$p < 0.01$, ###$p < 0.001$, ####$p < 0.0001$. #$p<0.05$, ##$p<0.01$, ###$p<0.001$ indicate comparisons between two indicated groups.

three weeks post-infection and then decreased at four weeks post-infection. Fig 5B shows that the expression of IL-5 in mice significantly increased at three weeks post-infection and then decreased after four weeks (p<0.01). Our results indicated different characteristic patterns in the expression of type-1 and type-2 cytokines in mouse and rat serum during the entire infection course.

## Discussion

Recent studies have shown that climate warming has led to the expansion or shift in the range of *A. cantonensis*, causing it to spread northward from the equator to several non-endemic areas, including Europe and the continental United States, potentially causing devastating harm [44]. The diverse symptoms of angiostrongyliasis make clinical diagnosis and treatment difficult. New diagnostic modalities for detecting changes in brain pathologies are urgently required. In this study, we developed a cross-disciplinary research model that combines parasitology and nuclear imaging. We first used the nanoPET nuclear imaging technique to analyze brain inflammation resulting from angiostrongyliasis. Our results demonstrated different patterns of radioactive uptake in different brain regions of infected mice and rats. Significant F[18]-FDG accumulations were observed in the hippocampus, thalamus, cerebellum, and brain stem of mice after infection. However, rats did not demonstrate significant changes as mice did.

*A. cantonensis* L3 larvae are infective to rats, humans, and other susceptible hosts. L3 larvae are transported from the blood to the rat brain, and then they penetrate the CNS within 24 hours. The larvae continue to grow within the brain and reach the L4 stage on the sixth or seventh day after ingestion. By the time they become young subadults (L5), they depart from the brain and reach the heart and lungs, with the majority ultimately residing in the pulmonary artery [45]. Our results showed that worms were still located in the brain two and three weeks after *A. cantonensis* infection in rats; however, after four weeks, only a few worms remained in the brain, and most of them were found in the heart and pulmonary arteries. In mice infected with *A. cantonensis*, the worms reached the highest level in the brain at three weeks, decreased at four weeks, and did not enter the heart or lungs. *A. cantonensis* worm bodies can induce an intense inflammatory response with an eosinophilic reaction in the CNS, resulting in the classic presentation of eosinophilic meningitis. Dying parasites can cause more severe inflammation and affect other CNS functions [46]. We found a correlation between the duration of residence of the parasite in the host brain and the timing of the subsequent immune response.

FDG-PET/CT has become more relevant for diagnosing several infectious and inflammatory diseases, and particularly for therapy monitoring. Scientific evidence indicates that FDG-PET/CT is useful for diagnosing and evaluating therapy in various infectious and inflammatory diseases, such as vasculitis, sarcoidosis, and spondylodiscitis [32,33,47]. The results of our study showed that the uptake of [18]F-FDG in the brains of rodents infected with *A. cantonensis* was higher at three weeks post-infection than that in control mice and at other time points. However, the radioactivity uptake by separate brain regions exhibited a distinct picture in infected mice and rats. Radioactivity in mice increased mainly in the hippocampus, thalamus, cerebellum, and brainstem at three weeks post-infection, whereas radioactivity uptake in rats showed almost no significant increase after infection. The increased uptake of [18]F-FDG in the brain of mice may have been caused by a metabolic burst of accumulated granulocytes, especially eosinophils, which use glucose as an energy source after activation [19,33]. A previous study revealed that the [18]F-FDG uptake rate could serve as a biomarker of eosinophilic inflammation and local lung function in asthma [47]. Therefore, the infiltration of eosinophils into separate brain regions in different host species may explain the increased radioactivity uptake in different brain regions in mice and rats. Additionally, our data revealed the

migration of worm burden from the brain to pulmonary arteries in rats within three to four weeks post-infection. Whether the migration and departure of parasitic larvae in the rat brain led to a decrease in their immune responses requires further investigation.

Histopathological examination revealed that immune and inflammatory cells increased in the region near the limbic system (midbrain, hippocampus, and thalamus) in mice infected with *A. cantonensis* at two weeks post-infection, **and then peaked at the fourth week.** On the other hand, inflammatory infiltration was not very evident in the infected rats during the infection. A previous study explored the difference in brain inflammation reactions between mice and rats infected with *A. cantonensis*, and found that activated microglia secrete several pro-inflammatory cytokines and induce a major role in eosinophil chemotaxis and more serious inflammation and neuronal damage in the CNS of infected mice [22]. It has been suggested that microglia play a major role in eosinophil chemotaxis and are activated in angiostrongyliasis [22,24]. Our immunohistochemical staining data demonstrated that microglia were enhanced in the brain regions surrounding the limbic system in infected mice, especially after three to four weeks of infection, which was accompanied by a substantial enhancement in inflammation that manifested in the hippocampal periphery. However, microglial activation in infected rats was not significant, except in the subarachnoid brain tissues. In contrast, eosinophil chemotaxis in infected mice changed remarkably from the cortical parenchyma to the regions underlying the limbic system as the infection progressed, especially in tissues between the hippocampus and thalamus. Conspicuous expression was noted within these regions starting from two weeks post-infection, which then increased sharply after three weeks, along with a significant accumulation of eosinophils in the aforementioned tissue by the fourth-week post-infection. Although YM-1 increase was observed in infected rats at three weeks post-infection, albeit to a much lesser degree than in mice, no further increase was observed after four weeks, except for a slight increase in the background of peripheral cortical tissues. These findings are consistent with those obtained from the $^{18}$F-FDG PET imaging experiments described previously. Microglia form a network of glial cells that are regularly distributed throughout the CNS. Previous studies have indicated that the total number of non-neuronal cells, such as microglia, in the limbic system is relatively high [48]. Microglial activation often precedes the response of any other cell type in the brain, and they respond not only to changes in the structural integrity of the brain but also to very subtle changes in the environment [49]. Microglial activation is also a major cause of inflammation and a key factor in the defense of neural parenchyma against infectious diseases, inflammation, ischemia, brain tumors, and neurodegeneration [50]. It is particularly noteworthy that Iba-1 increased significantly three weeks after infection in mice, demonstrating significant microglial activation. However, starting from two weeks after infection, before the increase in Iba-1, YM-1 expression increased significantly. Whether other factors are involved in eosinophilic chemotaxis at the site of infection before three weeks post-infection requires further exploration.

Based on the gene and protein expressions of Iba-1 and YM-1, Iba-1 expression was first enhanced in the second week after the worms entered the mouse brain, indicating that microglia, which are resident macrophages in the brain, were quickly activated. Microglia normally remain quiescent; however, the soluble antigen (sAg) released by *A. cantonensis* can immediately activate microglia and upregulate Iba-1 expression. Activation of microglia then skews the immune responses toward Th2, promoting the production of IgE and chemotaxis of eosinophils to combat parasitic infection [22,24,51]. **Due to penetration of parasite antigens into limbic system of brains in mice, activating microglia and subsequent activation of YM-1 eosinophil responses. Most microglia are distributed around the blood vessels of the limbic system of the brain, and evenly distributed in the parenchymal area of cerebral cortex. Therefore, the Iba-1 expression trend may be diluted in whole brain, while the periphery**

of the limbic system still shows a gradually and continuous increase trend of expression in infection. Previous studies have also reported that when the blood-brain barrier (BBB) is disrupted, Iba-1 will respond preferentially, activate and migrate to the injured site. Since *A. cantonensis* will easily break through the blood-brain barrier and enter the interior of the brain such as the limbic system and cerebellum in non-permissive host's brains. Our data further showed that Iba-1 up-regulation was maintained until four weeks post-infection [52–54]. At the same time, the eosinophil marker Ym-1 showed a significant and gradual increase in expression level from two to three weeks post-infection and reached a peak at four weeks post-infection. These findings indicate that *A. cantonensis* infection exacerbates inflammatory responses in the mouse brain, leading to activation of more microglia, inducing an increase in chemotaxis and eosinophil accumulation in the brain, and triggering a Th2-biased immune response feedback into an amplified cycle [55]. This result further validates our imaging findings that the uptakes of [18]F-FDG in the mouse brain occurred due to a massive influx of eosinophils and subsequent respiratory bursts. In contrast, the expression of Iba-1 and YM-1 in rat brains did not increase significantly during the entire infection period, and only YM-1 protein showed slight differences at four weeks after infection compared to the first two weeks. This result is also consistent with our [18]F-FDG imaging results, which showed that *A. cantonensis* does not produce a strong immune-inflammatory response in the brain of the permissive host.

A previous study reported that eosinophil chemotactic factors in angiostrongyliasis are mainly released from activated microglia in mice and rats and that these different complicated chemokine networks mediate the recruitment of eosinophils between permissive and non-permissive hosts during *A. cantonensis* infection [56]. Analysis of CBA results revealed that the expression of TNF in the serum of mice increased concomitantly with the prolongation of infection time, indicating an inflammatory response in mice that intensified with an increase in the infection duration. However, the upregulation of TNF was not observed in rats. This discrepancy implies that TNF may instigate processes of apoptosis and necroptosis in response to *A. cantonensis* infection in mice, but not in rats [57,58]. Furthermore, unexpectedly, IFN-γ and IL-4 in rat serum were significantly higher than that in mouse serum at two weeks post-infection, while rat serum IL-4 increased again at the fourth-week post-infection. Although the levels of IFN-γ and IL-4 in mouse serum showed a gradually increasing trend, there was no significant difference. This result was significantly different from the immune responses observed in the brains of rats and mice. Previous studies on cytokine levels in non-permissive hosts have shown that the reactivity of the immune system's IL-2, IL-4, and IL-5 significantly increased two and three weeks after *A. cantonensis* infection [59,60]. Our recent studies also showed that the systemic and regional immune responses of *A. cantonensis*-infected hosts are inconsistent and vary depending on the host strain [38]. Notably, the expression of IL5 in the serum of mice at the third week post-infection was consistent with the speculation that *A. cantonensis* infection induces a large number of eosinophils to proliferate and chemotaxis into the brain, subsequently triggering a strong inflammatory response. Previous studies have also reported that the levels of IgG and IgE antibodies against excretory and secretory antigens in the serum of BALB/c mice increased significantly starting from the third week after infection and continued until the fifth week, and their eosinophil counts also increased from the third week. However, the increase in eosinophil count led to the worsening of brain inflammation and brain tissue damage, and this time point was consistent with the death of BALB/c mice [38,61,62].

In summary, we used an *in vivo* [18]F-FDG /PET imaging model to evaluate live neuroinflammatory pathological changes in the brains of *A. cantonensis*-infected mice and rats. The uptake of [18]F-FDG increased in the cerebellum, brainstem, and limbic system of mice, whereas uptake in the rat brain was not significant. We further demonstrated that the uptake of [18]F-FDG in

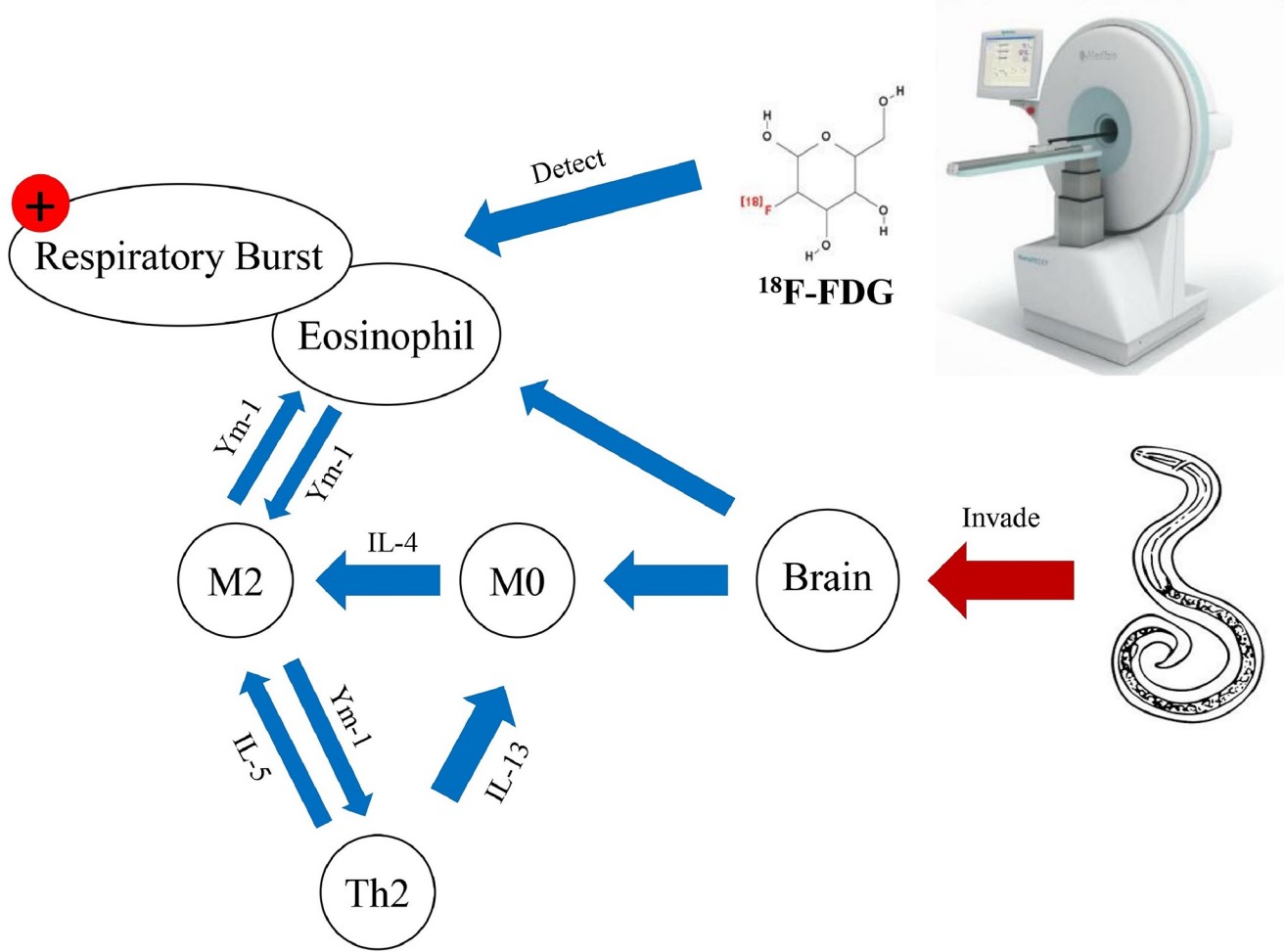

**Fig 6. Summary data map of this study.**

the host brain may be attributed to the accumulation of a large number of immune cells, especially the metabolic burst of activated eosinophils, which are attracted to and induced by activated microglia in the brain (Fig 6). In addition, although the results obtained in this study support our inference, the ICR mice used in this study were less susceptible to *A. cantonensis* and the degree of inflammatory response induced by it may not have been sufficient to simulate actual clinical conditions. At the same time, parts of the whole brains of mice and rats were used in the experiment to detect genes and proteins, which made it difficult to accurately reveal the areas of significant brain inflammation. Therefore, in future studies, we will quantitatively measure [18]F-FDG uptake in the brains of BALB/c mice susceptible to *A. cantonensis* infection. Different brain regions of the host will also be isolated for histopathological and protein examination of inflammation to confirm their relationship with radioactivity values. Through this study, we developed a specific radiological medical imaging analysis model for angiostrongyliasis for clinical diagnosis and treatment of important endemic zoonotic parasitic diseases worldwide. The development of nuclear medicine imaging techniques for the detection of parasitic diseases will greatly benefit the development and management of medical treatments.

## Supporting information

**S1 Data. Raw data for Figs 1 and 2.**
(RAR)

**S2 Data. Raw data for Fig 3.**
(RAR)

**S3 Data. Raw data for Figs 4 and 5.**
(RAR)

## Author Contributions

**Conceptualization:** Kang-wei Chang, Lian-Chen Wang, Po-Ching Cheng.

**Data curation:** Kang-wei Chang, Hung-Yang Wang, Tzu-Yuan Lin.

**Formal analysis:** Kang-wei Chang, Hung-Yang Wang, Edwin En-Te Hwu.

**Funding acquisition:** Edwin En-Te Hwu, Po-Ching Cheng.

**Investigation:** Kang-wei Chang, Lian-Chen Wang.

**Methodology:** Kang-wei Chang, Hung-Yang Wang, Tzu-Yuan Lin.

**Project administration:** Po-Ching Cheng.

**Resources:** Kang-wei Chang, Lian-Chen Wang, Po-Ching Cheng.

**Supervision:** Lian-Chen Wang, Po-Ching Cheng.

**Validation:** Tzu-Yuan Lin, Edwin En-Te Hwu.

**Visualization:** Hung-Yang Wang.

**Writing – original draft:** Kang-wei Chang, Hung-Yang Wang, Po-Ching Cheng.

**Writing – review & editing:** Lian-Chen Wang, Tzu-Yuan Lin, Edwin En-Te Hwu, Po-Ching Cheng.

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
