## [Decision Letter · Decision Letter 0]

7 Feb 2024

Dear Dr. Cheng,

Thank you very much for submitting your manuscript "Inflammatory and immunopathological differences in brains of permissive and non-permissive hosts with Angiostrongylus cantonensis infection can be identified using 18F/FDG/PET-imaging" for consideration at PLOS Neglected Tropical Diseases. As with all papers reviewed by the journal, your manuscript was reviewed by members of the editorial board and by several independent reviewers. In light of the reviews (below this email), we would like to invite the resubmission of a significantly-revised version that takes into account the reviewers' comments. 

We cannot make any decision about publication until we have seen the revised manuscript and your response to the reviewers' comments. Your revised manuscript is also likely to be sent to reviewers for further evaluation.

Sincerely,

Alessandra Morassutti

Academic Editor

Uriel Koziol

Section Editor

Reviewer's Responses to Questions

**Key Review Criteria Required for Acceptance?**

**Methods**

-Are the objectives of the study clearly articulated with a clear testable hypothesis stated?

-Is the study design appropriate to address the stated objectives?

-Is the population clearly described and appropriate for the hypothesis being tested?

-Is the sample size sufficient to ensure adequate power to address the hypothesis being tested?

-Were correct statistical analysis used to support conclusions?

-Are there concerns about ethical or regulatory requirements being met?

Reviewer #1: 1. The author mentioned in the text that ICR mice and SD rats were infected respectively with 50 and 100 L3. How is the number of infected larvae decided, and how the definite infected numbers are sure?

2. In the ELISA experiment, only mice sera were detected for IL5 without rats? please explain.

Reviewer #2: In the Methodology section, ensure consistency by describing actions in the past tense, as opposed to the predominantly future tense used by the authors. Please revise accordingly.

**Results**

-Does the analysis presented match the analysis plan?

-Are the results clearly and completely presented?

-Are the figures (Tables, Images) of sufficient quality for clarity?

Reviewer #1: 1. The FDG PET mouse results in Figure 1A reveal that there are obvious signals in the cerebellum and brainstem at the second week, but the data in Figure 1C only shows a value increase in the cerebellum. Why are there such differences between the cerebellum and brainstem? Please explain clearly how to distinguish them.

2. Fig 3A show Iba-1 in IHC gradually increase with the time of infection and peak at the 4th week; however, the expression of Iba-1 gene and protein in qPCR and WB increase significantly in the second week than the 4th week? Why they show different?

Reviewer #2: 1. In the text and Table 1 (p.15), the authors exclusively described the worm burden in the brain of ICR mice. For enhanced clarity and understanding, it would be beneficial to explicitly mention, for example, that worms were not detected in the hearts of ICR mice. (Note: The authors touch upon this point in the Discussion section)

2. Considering that eosinophils play a pivotal role as key effector inflammatory cells in cerebral Angiostrongyliasis, the authors should quantitatively assess the extent of eosinophil infiltration in different brain regions in correlation with YM-1 expression, rather than relying on descriptive wording. Similarly, a quantitative or semi-quantitative grading (i.e., 0, +, ++, +++) should be applied to assess microglia. If a direct count is challenging, a semi-quantitative approach would be acceptable.

3. To complement the aforementioned suggestion, in Fig. 2, the histopathology of eosinophil and microglia infiltration should be highlighted either as insets or as separate panels. For example, Fig. 2A could represent the current figures, while Fig. 2B could focus specifically on eosinophil and microglia infiltration with high magification.

**Conclusions**

-Are the conclusions supported by the data presented?

-Are the limitations of analysis clearly described?

-Do the authors discuss how these data can be helpful to advance our understanding of the topic under study?

-Is public health relevance addressed?

Reviewer #1: yes

Reviewer #2: NA

**Editorial and Data Presentation Modifications?**

Reviewer #1: major revision

Reviewer #2: (No Response)

**Summary and General Comments**

Reviewer #1: 1. The author mentioned in the text that ICR mice and SD rats were infected respectively with 50 and 100 L3. How is the number of infected larvae decided, and how the definite infected numbers are sure?

2. The FDG PET mouse results in Figure 1A reveal that there are obvious signals in the cerebellum and brainstem at the second week, but the data in Figure 1C only shows a value increase in the cerebellum. Why are there such differences between the cerebellum and brainstem? Please explain clearly how to distinguish them.

3. Fig 3A show Iba-1 in IHC gradually increase with the time of infection and peak at the 4th week; however, the expression of Iba-1 gene and protein in qPCR and WB increase significantly in the second week than the 4th week? Why they show different?

4. In the ELISA experiment, only mice sera were detected for IL5 without rats? please explain.

Reviewer #2: The authors present a comprehensive study on the use of 18F-FDG PET imaging to diagnose Angiostrongylus cantonensis infection in the brains of both permissive (rats) and non-permissive (mice) hosts, examining inflammatory responses over a 4-week period. Their findings reveal significant 18F-FDG uptakes in the cerebellum, brainstem, and limbic system of mice, contrasting with the absence of such uptake in rat brains. Immunohistochemical staining and western blotting demonstrate a substantial increase in macrophage-derived microglia (Iba-1 positive) and the eosinophil chemotactic factor, YM-1, in mice brains, with a less pronounced effect observed in rats. Serum levels of systemic proinflammatory cytokines (TNF, IFN-γ, IL-2) and anti-inflammatory cytokines (Th2, IL4, IL-5) differ between mice and rats, though conclusions are not yet definitive. The authors propose that 18F-FDG uptake in the host brain may be attributed to the accumulation of immune cells, particularly the metabolic burst of activated eosinophils, attracted and induced by activated microglia in the brain. While the overall experimental design and methodology are straightforward, there are concerns regarding the results and the presentation of findings in English.

The specific comments are listed in different sections above.

Comments from Reviewer 3:

This study is the first to examine the brain of rats and mice infected by Angiostrongylus cantonensis using PET imaging, combined with cytokine profiling and histology and parasitological assessment. The overarching idea – is to get a better understanding of this disease in human patients by comparing the response seen in the permissive (definitive) host – the rat, with that in an accidental host – the mouse. Lots of elegant and complicated imaging and histology and cytokine profiling is done in these 2 rodent animal models.

But there are some inherent weaknesses in the study, The authors look at the brain, but not at the spinal cord, and some of the most severe pathological processes occur in the spinal cord. The same inoculum is given to both rats and mice – but mice weigh maybe 10 grams, while rats weigh 300-1000 mg. And 50 large is a very heavy inoculum for either species, but especially in the mouse. Yet the authors do not common *except for a passing mention in the discussion) that the animals did not develop neurological; signs. However, we do not know how carefully they looked, and CSF was not collected from any animal (it’s pretty routine to collect CSF from rats).

If I had been doing these experiments, i think i would have used the guinea pig instead of the mouse – so that the permissive and non-permissive hosts were roughly the same side. And I would have done detailed neurological assessment of all infected animals, and undertaken PET imaging of the spinal cord as well as the brain and done histological assessment and cytokine profiling in the cord as well as different areas in the brain. And as the authors no doubt have access to excellent imaging, I would have been interested in high field MRI scans at the same time as the PET scanning.

I am not convinced the patterns of inflammation seen in the mice would be all that different from a model of cryptococcosis, or a model of bacterial meningoencephalitis – so I think the notion that this will help diagnose human cases of rat lungworm disease is a bit of a stretch. And with the new Arcan qPCR that has been developed, combined with the new antigen detecting systems, the diagnosis of neural angiostrongyliasis is now both extremely sensitive and highly specific, so I really doubt PET will have a lot to add. Having said that – it is a shame the authors did not include a study from an affected human and dog with Angiostrongylus infection, as it might be that the spatial distribution of 8F/FDG/PET signal might be more characteristic in a MUCH LARGER brain and spinal cord.

The authors also need to revise the grammatical structure of the manuscript – as their tenses get all screwed up in the Method where they use the future tense. I cannot really assess the histological sections as they are reproduced small, and do not have much spatial recognition.

PLOS authors have the option to publish the peer review history of their article (what does this mean?). If published, this will include your full peer review and any attached files.

Reviewer #1: No

Reviewer #2: No
---

## [Decision Letter · Decision Letter 1]

2 May 2024

Dear Dr Po-Ching Cheng,We are pleased to inform you that your manuscript 'Inflammatory and immunopathological differences in brains of permissive and non-permissive hosts with Angiostrongylus cantonensis infection can be identified using 18F/FDG/PET-imaging' has been provisionally accepted for publication in PLOS Neglected Tropical Diseases.

Best regards,

Alessandra Morassutti, PhD

Academic Editor

Uriel Koziol

Section Editor

Reviewer's Responses to Questions

**Key Review Criteria Required for Acceptance?**

**Methods**

-Are the objectives of the study clearly articulated with a clear testable hypothesis stated?

-Is the study design appropriate to address the stated objectives?

-Is the population clearly described and appropriate for the hypothesis being tested?

-Is the sample size sufficient to ensure adequate power to address the hypothesis being tested?

-Were correct statistical analysis used to support conclusions?

-Are there concerns about ethical or regulatory requirements being met?

Reviewer #1: yes

Reviewer #2: OK

**Results**

-Does the analysis presented match the analysis plan?

-Are the results clearly and completely presented?

-Are the figures (Tables, Images) of sufficient quality for clarity?

Reviewer #1: yes

Reviewer #2: OK

**Conclusions**

-Are the conclusions supported by the data presented?

-Are the limitations of analysis clearly described?

-Do the authors discuss how these data can be helpful to advance our understanding of the topic under study?

-Is public health relevance addressed?

Reviewer #1: yes

Reviewer #2: OK

**Editorial and Data Presentation Modifications?**

Reviewer #1: Accept

Reviewer #2: NA

**Summary and General Comments**

Reviewer #1: no comments

Reviewer #2: The authors have addressed all my comments with satisfaction. It is now suitable for publication.

PLOS authors have the option to publish the peer review history of their article (what does this mean?). If published, this will include your full peer review and any attached files.

Reviewer #1: No

Reviewer #2: No

---

## [Editor Report · Acceptance letter]

21 May 2024

Dear Dr. Cheng,

We are delighted to inform you that your manuscript, "Inflammatory and immunopathological differences in brains of permissive and non-permissive hosts with Angiostrongylus cantonensis infection can be identified using 18F/FDG/PET-imaging," has been formally accepted for publication in PLOS Neglected Tropical Diseases.

Best regards,

Shaden Kamhawi

co-Editor-in-Chief

Paul Brindley

co-Editor-in-Chief
